# Haloalkalitolerant Fungi from Sediments of the Big Tambukan Saline Lake (Northern Caucasus): Diversity and Antimicrobial Potential

**DOI:** 10.3390/microorganisms11102587

**Published:** 2023-10-19

**Authors:** Marina L. Georgieva, Elena N. Bilanenko, Valeria B. Ponizovskaya, Lyudmila Y. Kokaeva, Anton A. Georgiev, Tatiana A. Efimenko, Natalia N. Markelova, Anastasia E. Kuvarina, Vera S. Sadykova

**Affiliations:** 1Gause Institute of New Antibiotics, St. Bolshaya Pirogovskaya 11, 119021 Moscow, Russia; efimen@inbox.ru (T.A.E.); nathanmrk82@gmail.com (N.N.M.); nastena.lysenko@mail.ru (A.E.K.); 2Faculty of Biology, Lomonosov Moscow State University, 1-12 Leninskie Gory, 119234 Moscow, Russia; e_bilanenko@mail.ru (E.N.B.); v.ponizovskaya@gmail.com (V.B.P.); kokaeval@gmail.com (L.Y.K.); semga2001@yandex.ru (A.A.G.); 3Faculty of Soil Sciences, Lomonosov Moscow State University, 1-12 Leninskie Gory, 119991 Moscow, Russia

**Keywords:** alkalitolerant fungi, halotolerant fungi, *Emericellopsis*, antimicrobial activity, saline lake

## Abstract

We have performed a characterization of cultivated haloalkalitolerant fungi from the sediments of Big Tambukan Lake in order to assess their biodiversity and antimicrobial activity. This saline, slightly alkaline lake is known as a source of therapeutic sulfide mud used in sanatoria of the Caucasian Mineral Waters, Russia. Though data on bacteria and algae observed in this lake are available in the literature, data on fungi adapted to the conditions of the lake are lacking. The diversity of haloalkalitolerant fungi was low and represented by ascomycetes of the genera *Acremonium*, *Alternaria*, *Aspergillus*, *Chordomyces*, *Emericellopsis*, *Fusarium*, *Gibellulopsis*, *Myriodontium*, *Penicillium*, and *Pseudeurotium*. Most of the fungi were characterized by moderate alkaline resistance, and they tolerated NaCl concentrations up to 10% *w*/*v*. The analysis of the antimicrobial activity of fungi showed that 87.5% of all strains were active against *Bacillus subtilis*, and 39.6% were also determined to be effective against *Escherichia coli*. The majority of the strains were also active against *Aspergillus niger* and *Candida albicans*, about 66.7% and 62.5%, respectively. These studies indicate, for the first time, the presence of polyextremotolerant fungi in the sediments of Big Tambukan Lake, which probably reflects their involvement in the formation of therapeutic muds.

## 1. Introduction

High salt concentrations and alkaline pH values in natural habitats yield the emergence of specific communities of prokaryotic and eukaryotic organisms, with a pronounced dominance of a small number of adapted species [1,2,3,4,5,6]. Every part of the community plays its own role and is strongly connected to the other parts [7,8,9]. Studying cultivated filamentous fungi and their involvement in salt- and alkalitolerant organisms’ communities is of great importance for the examination of fungal taxonomic diversity, physiological characteristics, and the evaluation of antimicrobial potential [10,11,12,13,14]. Investigations of fungi from the coasts of lakes with different degrees of salinity and alkalinity demonstrated that obligate alkaliphilic and halophilic fungi are quite rare and more often confined to specific extreme environments [15,16,17,18], while haloalkalitolerant fungi, i.e., growing at high pH values and salt concentrations but not requiring them for their development, are widespread [14,17,19,20]. It is surprising that some obligate alkaliphilic fungi can be found at the bottom of lakes with neutral chloride salinization, like Lake Baskunchak [21]. The explanation of this phenomenon might be based on the fact that cyanobacterial films and mats, often found in salt lakes, create localized alkaline conditions as a result of photosynthesis and autotrophic sulfate reduction [17]. These films and mats then decompose to form bottom sediments with therapeutic effects (therapeutic muds).

Stress-tolerant fungi are a crucial functional component of the degrader community in an environment with increased salinity and alkalinity [8,9]. Adaptation to one stress factor is usually coupled with adaptation to other factors, i.e., polystress. For example, high salt concentrations cause a lack of free water and are usually associated with periodically high temperatures. Fungi living under such conditions appear to be polyextremophiles [2,3,22,23]. These unique conditions lead to the production of new metabolites (enzymes, antibiotics, etc.) with great biotechnological potential. New antimicrobial peptides named emericellipsins A–E and hydrophobins with antibacterial and antifungal activities were recently discovered to be produced by the alkaliphilic fungi *Emericellopsis alkalina* and *Sodiomyces alkalinus* [24,25]. These findings demonstrate the significance of conducting antimicrobial activity studies on the fungi present in these extreme environments.

Big Tambukan Lake, a saline and slightly alkaline drainless lake located in southern Russia, is the main source of therapeutic mud used for treatments in the sanatoria of the Caucasian Mineral Waters [26,27,28,29,30]. The use of muds (peloids) in medical therapy dates back to ancient times, mainly in the treatment of locomotor system pathologies and dermatology [30,31,32,33,34,35,36]. The mud consists of various organic and inorganic matter and their mixture [37,38,39]. Microorganisms play a distinct role in the maturation process of mud mixtures: together with prokaryotes, there are also reports of algae (diatoms, *Dunaliella*, *Vaucheria*) [11,40,41,42,43,44,45]. The Russian microbiologist B.L. Isachenko was the first to study the genesis of therapeutic mud from Big Tambukan Lake and established the role of sulfur-cycle bacteria in this process [44]. The salinity of the brine lake over the last 100 years has decreased from 481.1 (in 1930) to 21.4 (in 1997) g/L, but the chloride–sulfate magnesium–sodium composition of the brine has been preserved. The water pH values vary from 8 to 9 [26,28]. Because of this, significant changes occurred in the biota of the reservoir and influenced the formation of therapeutic mud [29,46]. No information could be found in the literature concerning mycological studies of the sediments or muds of this lake. The high salinity and alkaline pH values of the water of this lake caused the focus to be placed on haloalkalitolerant fungi, among which it is possible to find producers of valuable antimicrobial compounds.

The objective of the present study is to provide a description of haloalkalitolerant fungi (based on morphological characters and molecular–genetic analysis) isolated from sediments of Big Tambukan Lake and their ecophysiological parameters (growth at different pH values, NaCl concentrations, and antimicrobial activity). To our knowledge, this is the first report of the presence of haloalkalitolerant fungi in Big Tambukan Lake and their antimicrobial activities.

## 2. Materials and Methods

### 2.1. Samples

Ten samples were collected in August 2018 on the north coast of Big Tambukan Lake (hereinafter Tambukan), Northern Caucasus, Russia (43°57′52.6″ N 43°09′56.8″ E) (Figure 1). The lake’s surface area is 1.87 square kilometers, and its depth is about 3 m. During sampling, the salinity of the lake water was 28 g/L, and its pH level was 8.2. The samples were the top 0–5 cm layer of underwater sediments collected along the coastline. They included clay, sand, silt, mudflat soils, and cyanobacterial mats’ remains. The pH of the samples was 7.5–8.0. The shores of the lake are overgrown with dense vegetation, and among trees, oak and beech dominate. From the bottom of the central part of the lake, mud is derived.

### 2.2. Culture Media

The selective isolation of haloalkalitolerant fungi was carried out using an alkaline agar medium (AA, pH 10) prepared from malt–yeast extract and carbonate–bicarbonate buffer and containing the following salts: NaCl—6 g/L; KNO_3_—1 g/L; K_2_HPO_4_—1 g/L. Medium preparation is described in detail by Grum-Grzhimaylo et al. [47]. Bacterial growth was inhibited by rifampicin (2 g/L), which was the most efficient antibiotic among 21 preparations tested on AA [16]. Oatmeal agar (OA), Czapek-Dox agar (CZA), and malt–yeast agar (MYA) were used as standard media to culture the obtained micromycetes.

Buffer choices for generating pH values were made according to Grum-Grzhimaylo et al. [16]: Na_2_HPO_4_/NaH_2_PO_4_ system for pH 8 and Na_2_CO_3_/NaHCO_3_ system for pH 10. Thus, media buffered at pH 8.0 and 10.1 consisted of two components: the buffer component and MYA as the nutrient component. These components were autoclaved separately and then mixed afterward, making up the complete medium. Buffered medium preparation is described in detail by Grum-Grzhimaylo et al. [47].

The reaction of the isolates to the various NaCl concentrations on MYA medium containing either 0%, 2.5%, 5%, 7.5%, or 10% NaCl (*w*/*v*) was studied.

### 2.3. Isolation Techniques

Inoculation was performed using a soil-lump technique, i.e., via the distribution of soil lumps on the surfaces of Petri plates filled with AA. Inoculated plates were wrapped in Parafilm and incubated at 22 °C. After 10–21 days of incubation, the colonies were enumerated, and the retrieved fungal isolates were purified and pre-identified by morphological characters. Forty-eight isolates with different colony morphologies or, in the case of similar phenotypic variants, belonging to different sediment samples were selected for further studies. The 48 newly acquired strains were deposited at the Collection of Extremophilic Fungi of the Department of Mycology and Algology of Moscow State University (MSU) (Russia).

The fungal communities were analyzed according to the frequency of occurrence and the abundance of isolated species. The frequency of the occurrence of a particular species was defined as the percent composition of the number of samples in which a particular species occurred relative to the total number of all samples obtained. The abundance of a particular species was defined as the percent composition of the colony-forming units (CFU) of a particular species relative to the total CFU of all species obtained.

### 2.4. Morphological Identification of Fungal Isolates

The macromorphology of colonies and their micromorphological characters were observed for 14-day-old colonies grown on AA, MYA, OA, and CZA. The study of fungal morphology was investigated using a light microscope (LM) and scanning electron microscope (SEM). LM observations were performed on a Leica DM 2500 microscope equipped with a DFC 495 camera. SEM specimens were prepared by fixing them in approximately 4% formaldehyde or 2.5% glutaraldehyde, followed by washing and dehydration steps (20 min each) with a series of ethanol (30%, 50%, 70%, and 96% concentrations) and acetone before drying at the critical point in CO_2_, followed by metal sputter coating. SEM observations were performed using JSM-6380LA (JEOL, Tokyo, Japan) and Quattro S (Thermo Scientific, Brno, Czechia) microscopes. Images were acquired and elaborated using MicroCapture (version 1.0) software.

Manuals, identification guides, and original articles on fungal taxonomy were used for identification. Names and systematic positions of fungi are given in accordance with MycoID [48], Westerdijk Fungal Biodiversity Institute [49], and The National Center for Biotechnology Information (GenBank) [50].

### 2.5. Molecular Identification of Fungal Isolates

Molecular identification of fungal strains was performed by PCR amplification of the rDNA internal transcribed spacer region (ITS) using the primers ITS1F (TCCGTAGGTGAACTTGCG) and ITS4R (TCCTCCGCTTATTGATATGC) [51]. Genomic DNA was extracted using a genomic DNA isolation kit (LLC Biolabmix, Novosibirsk, Russia) as per the manufacturer’s protocol. The 50 µL PCR reaction mixture consisted of 25 µL of BioMaster HS-Taq PCR-Spec (2×) reaction mix (LLC Biolabmix, Novosibirsk, Russia), 50 mmol MgCl_2_, 0.1 nmol of each primer, 1–100 ng of isolated DNA, and nuclease-free water. PCR was carried out on a Thermal Cycler 2720 (Applied Biosystems, Foster City, CA, USA) with the following protocol: (1) 95 °C for 5 min, (2) 33 cycles of denaturation at 95 °C for 1 min, annealing at 51 °C for 1 min, extension at 72 °C for 1 min, and (3) final extension at 72 °C for 7 min. DNA fragments were subjected to Sanger sequencing on an Applied Biosystems 3500 Series Genetic Analyzer (Applied Biosystems, Beverly, MA, USA) using the primers ITS1F, ITS2R (GCTGCGTTCTTCATCGATGC), ITS3F (GCATCGATGAAGAACGCAGC), and ITS4R. Additionally, beta-tubulin (*b-tub*) gene amplification was performed for *Emericellopsis* species using the βt2a (GGTAACCAAATCGGTGCTGCTTTC) and βt2b (ACCCTCAGTGTAGTGACCCTTGGC) primers.

### 2.6. Phylogenetic Analysis

We generated an ITS alignment to fit all available sequences that could be reliably aligned with our own. To this end, we retrieved sequences from GenBank (Appendix A). Alignments were calculated through the MAFFT 7.429 online server [52] using the L-INS-I strategy. *Emericellopsis* isolates were identified via the sequencing of two gene loci, ITS rDNA and *b-tub*. The nucleotide sequences were submitted to GenBank [50] with their assigned accession numbers (Table 1).

For the taxonomic identification of species, phylogenetic analysis was carried out using the maximum likelihood (ML). Several phylogenetic trees were constructed at different taxonomic levels to achieve a more precise determination of the taxonomic positions of individual isolates. Nucleotide sequences for these constructions were retrieved from relevant fungal group articles and the NCBI database [50]. ML was run on RAxML-NG1.1.0 [53] with one thousand bootstrap replicates. For ML analyses, IQ-TREE 1.2.2 [54] with the best-fitted model option was used. Bootstrapping was performed using the standard nonparametric bootstrap algorithm with the number of replicates set to 1000. Support for nodes is indicated with bootstrap values ≥ 70.

### 2.7. Growth at Different pH Values and Salt Concentrations

To elucidate the pH preferences of the strains, three media were used. They were MYA (pH 6.1) and two MYA-based media buffered at pH levels 8.0 and 10.1. Grum-Grzhimaylo with co-authors [16] pointed out that sometimes, the presence of extra Na^+^ in small amounts in the medium could significantly enhance fungal growth in conditions of high pH stress. In this way, to reveal the fungal growth type more completely, in addition to the listed media for growth experiments, we used AA medium (pH 10) containing 0.6% NaCl (*w*/*v*).

To reveal halotolerant properties, we studied the reaction of the isolates to the various NaCl concentrations on the MYA medium.

Petri dishes were inoculated at the center, then sealed thoroughly with Parafilm to avoid desiccation, and put at 25 °C in the dark. The growth expansion was followed for 2–4 weeks, depending on the strain. Every two or three days, we measured the diameters of fungal colonies in two perpendicular directions. To determine the linear growth of the strains, a linear regression model was used. The regression coefficient numerically corresponded to the linear growth rate of the colony. The confidence intervals (CIs) for the means were calculated according to Student’s t-distribution (*n* = 4, α = 0.05) and plotted on the graphs.

### 2.8. Antimicrobial Assays

The strains of fungi were grown on CZA, MYA, and AA. Our prior research demonstrated that an alkaline medium stimulates the synthesis of peptaibols with antimicrobial activity [55,56,57]. The detection of the peptaibol emericellipsin A in the crude extracts from *Emericellopsis* strains was made by HPLC and MALDI-TOF MS spectra in accordance with previously obtained data [24,58].

The antimicrobial activity of the strains was measured by the disk diffusion method on test cultures of mycelial and yeast microscopic fungi and on bacteria from the collection of cultures of the Gause Institute of New Antibiotics (Moscow, Russia). Opportunistic mold and yeast test cultures of the fungal species *Aspergillus niger* INA 00760, *Candida albicans* ATCC 2091, and test cultures of Gram-positive strains such as *Bacillus subtilis* ATCC 6633 and Gram-negative bacteria such as *Escherichia coli* ATCC 25922 were used.

Disks with a 6 mm diameter containing 40 µL of the culture broth were deposited on agar plates. The diameters of the inhibition zones were measured after 24 h at 28 °C. Amphotericin B (for fungi) and ampicillin (for bacteria) solutions (Sigma-Aldrich, St. Louis, MO, USA) were used as positive controls. Cultures were considered to be highly active if the zone of growth retardation of the test organism was 20 mm or more; moderately active cultures had a growth retardation zone of 10–20 mm, and weakly active cultures had a zone of less than 10 mm (disk diameter 6 mm). The test culture of *B. subtilis* ATCC 6633 was grown on Gause medium no. 2 with the following composition (g/L): 2.5 tryptone (or 30 mL Hottinger broth), 5 peptone, 5 sodium chloride, 10 glucose. *E. coli* ATCC 25922 was grown on lysogen-broth LB medium (tryptone soy agar). Cultures of the fungi *A. niger* INA 00760 and *C. albicans* ATCC 2091 were grown on CZA. The cultures were preliminarily grown in test tubes with nutrient agar slants, after which their cells were suspended in saline to a turbidity of 0.5 according to the McFarland standard HiMedia R092R Tube, Maharashtra, India (1.5 × 10^8^ CFU/mL) and were used within 15 min. One-day cultures of bacteria and 5-day cultures of molds and yeasts were used.

## 3. Results

### 3.1. Analysis of Cultivated Haloalkalitolerant Fungi

From the sediments of Tambukan Lake, 272 fungal CFU were obtained. The retrieved isolates were represented by 10 genera belonging to four classes, i.e., Sordariomycetes, Dothideomycetes, Eurotiomycetes, and Leotiomycetes, all within the phylum Ascomycota. The largest number among all detected species belonged to the *Emericellopsis* genus. Other genera recorded with more than one species were *Aspergillus*, *Fusarium*, *Penicillium*, and *Gibellulopsis* (Table 2).

We detected the presence of fungi in all samples of sediments obtained (Appendix A). The number of isolated strains ranged from 6 to 37 per sample, reaching, on average, 27 strains per sample (Figure 2).

*Emericellopsis* was the most frequently occurring genus (100%) and was isolated from all of the sediment samples (Table 2). Other genera detected with a high frequency of occurrence were *Alternaria* (60%), *Aspergillus* (50%), *Penicillium* (50%), and *Fusarium* (40%). The rest of the detected genera occurred in one or two samples, and their occurrence was lower than 20%.

The abundance of the *Emericellopsis* genus was 66.2%, varying from 35% (sample no. 4) to 100% (sample no. 7) in a particular sample. Then, *Alternaria alternata* possessed a high abundance (10%). The maximum abundance was noted in sample no. 6 and amounted to 62.5%. High abundance was also noted for *Aspergillus* (8.1%), *Penicillium* (6.3%), and *Fusarium* (5.5%). The abundance of other isolates did not exceed 2%.

### 3.2. Molecular Identification, Phylogenetic Analyses, and Salt and Alkaline Preferences of the Obtained Isolates

Forty-eight isolates were analyzed. The vast majority of them were able to develop in an alkaline medium (i.e., pH 8), though they significantly reduced their growth rate in a strongly alkaline medium (i.e., pH 10), and thus, we treat them as alkalitolerants. *Pseudeurotium bakeri* (p47) was the single isolate obtained that could not tolerate pH 8. We further divide alkalitolerants according to their growth characteristics into strong, moderate, and weak [16]. Strong alkalitolerants grew at pH 8 considerably better than at pH 6 and showed little reduction in growth at pH 10. Moderate alkalitolerants showed better growth at pH 8 than at pH 6, or the same at both pH levels, and strongly reduced growth at pH 10. Weak alkalitolerants preferred a slightly acidic medium (pH 6), and their growth was hindered as pH increased.

Among obtained isolates, no halophiles were detected [18]. The tested isolates can be categorized into three groups—strong halotolerants that can grow optimally at 7.5% (*w*/*v*) NaCl and above, moderate halotolerants that can grow optimally at 2.5% and/or 5% (*w*/*v*) NaCl, and weak halotolerants that grow optimally in a medium without NaCl and whose growth rates decrease with increasing NaCl concentration.

#### 3.2.1. *Emericellopsis* spp. (Bionectriaceae, Hypocreales)

Fungi of the *Emericellopsis* genus dominated in the studied samples, possessing the highest frequency of occurrence and abundance (Table 2; Figure 2). ITS and *b-tub* sequences were generated for 20 isolates from sediments of Tambukan Lake, and these were aligned with 45 ITS and 36 *b-tub* sequences retrieved from GenBank, including 1 outgroup taxon (*Stanjemonium grisellum* CBS 655.79 T) (Table 1; Appendix A). *Emericellopsis* strains from the sediments of Tambukan Lake were clustered into three ecological clades, outlined previously by Zuccaro et al. [59], Grum-Grzhimaylo et al. [60], and Hagestad et al. [61]. Three strains were in the Soda clade; thirteen strains were in the Marine clade; and four strains were in the Terrestrial clade (Figure 3; Appendix A).

Three strains (p30, p36, p43) within the Soda clade were identified as *Emericellopsis alkalina*, as they clustered together with the ex-type strain of *E. alkalina* (CBS 127350 T) and other strains of *E. alkalina* (A117, A118, A119, and CBS 120049).

The majority of *Emericellopsis* isolates from the sediments of Tambukan Lake were classified within the Marine clade. Phylogenetic analysis based on the ITS locus showed that seven isolates (p37, p45, p32, p20, p39, p27, p22) grouped with the known ex-type strain of *E. maritima* (CBS 491.71 T), while the remaining six isolates (p38, p41, p40, p46, p21, p25) differed from the ex-type strain *E. maritima* and formed a distinct clade (Appendix A). Phylogenetic reconstructions based on two genes revealed considerable heterogeneity among isolates of this genus obtained from sediments of Tambukan Lake (Figure 3). Only strains p37 and p21 were found to be closely related to the ex-type strain of *E. maritima*, but they did not group together in a distinct clade. Instead, they formed sister groups with it. The other 11 isolates formed several distinct clades, strongly supported within their respective groups, and did not group with any known species of the genus. Morphologically, these isolates displayed similarities, forming Acremonium-like asexual sporulation-producing conidial heads on solitary, sometimes branched phialides, while the sexual stage was not observed.

Four strains, namely, p24, p26, p29, and p49, were placed in the Terrestrial clade. Strain p49 was identified as *Emericellopsis* sp., as it formed a separate sister clade in the phylogenetic analysis based on two loci, despite being grouped within the known ex-type strains of *E. terricola* (CBS 120.40 T) and *E. terricola* (CBS 229.59) in the ITS-based analysis (Figure 3; Appendix A). This isolate also lacked the teleomorph stage and only exhibited an Acremonium-like anamorph. Three strains (p24, p26, p29) within the Terrestrial clade were grouped together with *Emericellopsis fimetaria* in both the ITS-based and two-loci (ITS and *b-tub*) phylogenetic analyses (Figure 3; Appendix A). The isolates from the sediments of Tambukan Lake showed high support (100%) within this group. All three isolates formed abundant conidiation. Only the p24 isolate in culture (on MA and OA media) formed fruiting bodies typical of the *Emericellopsis* genus (Figure 4).

Ascomata developing in aerial and submerged mycelia were globose, non-ostiolate, and hyaline at the beginning and became black as ascospores matured, reaching 108–190 μm in diameter. Asci were eight-spored and (sub-) globose, with thin, evanescent, hyaline walls. Ascospores were unicellular and ellipsoid (6.5–8.0 × 3.8–5.4 μm) with 3–5 longitudinal wings (0.4–1.6 μm wide), most of which extended from pole to pole of the spore and which had an undulating or sometimes ragged margin. Colonies on OA at 25 °C after 14 days reached 40–42 mm in diameter and were felty, with reverse yellowish-white and, later, black zones appearing as a result of the formation of abundant ascomata. Colonies on MYA at 25 °C, after 14 days, were 34–36 mm in diameter and white, slightly pinkish, sometimes later darkening in the center due to the formation of ascomata (Appendix A). Conidiophores had repeated mesotonous verticillate branching, with branches diverging in acute angles. Stilbella-like synnemata did not form. Conidia were 3.1–3.7 × 1.7–1.9 mm, shorter than the ascospores.

Within the *Emericellopsis* genus, physiological characters, i.e., pH and NaCl concentration preferences, differed depending on the isolate (Figure 5; Appendix A).

All of the isolates from the Soda and the Marine clades preferred the weakly alkaline medium (pH 8) and were strong or moderate alkalitolerants that slightly or significantly reduced their growth rate in the strongly alkaline medium at pH 10. All of the obtained isolates from the Soda clade (p30, p36, and p43) were moderate alkalitolerants. In the Marine clade, ten isolates (p20, p22, p25, p27, p32, p37, p38, p39, p40, p45) revealed a strong alkalitolerant phenotype, and the remaining three isolates (p21, p41, p46) had a moderate one. In the Terrestrial clade, isolates were slow-growing, and all of them (p26, p29, p49, and p24) were moderate alkalitolerants with similarly high growth rates at both pH 6 and pH 8 (Figure 5a; Appendix A).

It is worth mentioning that at high alkaline ambient pH (10), the majority of isolates from the Marine clade (p20, p25, p27, p37, p41, p46) increased their growth rates on the AA medium compared to the medium without salt. Some of them (p20, p25, p27, p37), though, had restricted growth at pH 10 without NaCl, but in the presence of NaCl (0.6% *w*/*v*) in the AA medium, they demonstrated a growth rate comparable to that at pH 8, and thus, we assigned these isolates to the group of strong alkalitolerants. Other *Emericellopsis* isolates did not increase their growth rates on the AA medium (Figure 5a; Appendix A). Interestingly, at pH 10, all the *Emericellopsis* isolates demonstrated a prolonged adaptation period of about ten days, during which they grew extremely slowly. After that period, their growth rate was noticeably accelerated.

As for NaCl preferences, all of the isolates from the Soda clade (p30, p43, p36) were moderate halotolerants, as they preferred a medium containing NaCl, growing optimally at 2.5–5% NaCl. Then, eleven isolates from the Marine clade were moderate halotolerants (p20, p21, p22, p27, p32, p37, p39, p40, p41, p45, p46), with optimal growth at 2.5% and/or 5% NaCl and a decreased growth rate when the NaCl concentration exceeded 5%. The remaining two isolates (p25, p38) were strong halotolerants that grew optimally in a very wide range of NaCl concentrations, including 7.5% NaCl. In the Terrestrial clade, *E. fimetaria* isolates were moderate (p26 and p29) and strong (p24) halotolerants. They preferred media with both 0% and 2.5% NaCl (p26, p29) or grew on media containing 2.5–7.5% NaCl (p24). Finally, p49 was a moderate halotolerant that preferred media with both 0% and 2.5% NaCl (Figure 5b; Appendix A).

#### 3.2.2. *Alternaria alternata* (Pleosporaceae, Pleosporales)

ITS sequences were obtained for 2 isolates (p18, p44) and subsequently aligned with 13 sequences retrieved from GenBank, including 1 outgroup taxon (*Curvularia homomorpha* CBS 156.60 T) (Table 1; Appendix A). The phylogenetic analysis revealed that our *A. alternata* (p18 and p44) isolates clustered within a highly supported clade (99% ML) together with the ex-type strain of *A. alternata* (CBS 916.96 T) (Appendix A).

As for ecophysiological characters, *A. alternata* isolates were weak alkalitolerants, as they grew optimally at pH 6 and were able to grow at alkaline pH levels, though rather slowly. Then, these isolates were moderate halotolerants that preferred 0–2.5% NaCl (p18) or 0–5% NaCl (p44) (Figure 6; Appendix A).

#### 3.2.3. *Aspergillus* spp. (Aspergillaceae, Eurotiales)

The morphological analysis identified isolates p10, p11, p14, p16, p34, and p35 as *Aspergillus* species. Subsequently, ITS sequences were obtained for these six isolates and aligned with 22 sequences retrieved from GenBank, which included one outgroup taxon (*Aspergillus fumigatus* CBS 133.61 T) (Table 1; Appendix A). The phylogenetic analysis resulted in the clustering of the sequenced isolates into three distinct clades (Figure 7). The majority of isolates belonged to the *Flavipedes* section from the *Flavipedes* series (p35, p14) and *Spelaei* series (p34, p11). Additionally, two strains (p10, p16) were classified as members of the *Nidulantes* section from the *Versicolores* series (Figure 7).

Four *Aspergillus* isolates (i.e., p10, p14, p34, and p35) were moderate alkalitolerants, as their growth rate at pH 8 was higher than (p10, p14, p34) or the same as (p35) that at pH 6 and was strongly restricted at pH 10. The remaining two isolates (p11 and p16) were weak alkalitolerants, as they preferred pH 6 (Figure 8a).

In general, the *Aspergillus* isolates studied here preferred media containing NaCl at concentrations 7.5% and above, and thus, they were strong halotolerants. *Aspergillus* isolates from the *Versicolores* series showed high growth rates at 5–7.5% NaCl (p10) or at 2.5–10% NaCl (p16) and a severe growth decrease on a medium without added salt. Isolates among the *Spelaei* series (p11 and p34) and *Flavipedes* series (p14 and p35) grew well at all salt concentrations tested from 0 to 10% NaCl (Figure 8b).

#### 3.2.4. *Penicillium* spp. (Aspergillaceae, Eurotiales)

The morphological analysis identified isolates p1, p2, p4, p5, p6, p48, p301, and p302 as *Penicillium* species. Subsequently, ITS sequences were obtained for these 8 isolates and aligned with 21 sequences retrieved from GenBank, which included 1 outgroup taxon (*Aspergillus glaucus* CBS 516.65 T) (Table 1; Appendix A). The phylogenetic analysis resulted in the clustering of the sequenced isolates into three distinct clades (Figure 9). Most of the isolates (p2, p4, p5, p6, p301) formed a clade corresponding to *Penicillium* section *Chrysogena* series *Chrysogena*. Two isolates (p48, p302) clustered with the ex-type isolate of *P. bialowiezense* (CBS 227.28 T) and other species of the section *Brevicompacta* series *Brevicompacta*. The remaining isolate (p1) clustered with the ex-type isolate of *P. terrigenum* (CBS 127354 T) and other species of section *Citrina* series *Copticolarum*.

The vast majority of the studied *Penicillium* isolates, except for a single strain (p5), successfully developed on the slightly alkaline medium (pH 8). Six isolates were moderate alkalitolerants: their growth rate at pH 8 was a little higher than (p302) or the same as (p1, p4, p6, p48, p301) that at pH 6. Another two isolates (p2, p5) were weak alkalitolerants showing optimal growth at pH 6. At pH 10, the growth of all of the isolates was strongly reduced. The presence of extra salt in the AA medium did not have a visible effect on the growth pattern (Figure 10a).

In general, *Penicillium* isolates preferred NaCl-containing media and were moderate or strong halotolerants. Isolates among the *Chrysogena* series were moderate halotolerants, with optimal growth on media that included 5% NaCl (p2, p5), or strong halotolerants that grew well at all salt concentrations tested from 0 to 10% NaCl (p301) or preferred 5–10% NaCl (p4) or 5–7.5% NaCl (p6). A *Brevicompacta* series isolate (p48) preferred 5–10% NaCl and thus was strongly halotolerant, while the isolate p302 grew best at 5% NaCl and was moderate. Finally, *Penicillium* sp. series *Copticolarum* (p1) showed the highest growth rate at 2.5–5% NaCl and was a moderate halotolerant (Figure 10b).

#### 3.2.5. *Fusarium* spp. and *Acremonium* spp. (Hypocreales)

Based on the morphological analysis, isolates p7, p9, p12, p13, and p28 were identified as *Fusarium* species, whereas isolates p19 and p33 were identified as *Acremonium* species. Subsequently, ITS sequences were generated for the 7 isolates and aligned with 26 sequences retrieved from GenBank, including 1 outgroup taxon (*E. alkalina* CBS 127350 T) (Table 1; Appendix A). The phylogenetic analysis revealed that the isolates sequenced in this study clustered into four distinct clades (Figure 11). The *Fusarium* isolates formed three species complexes: Solani (p9, p13), Incarnatum-Equiseti (p7, p12), and Burgessii (p28). The *Acremonium* isolates (p19, p33) clustered together with the ex-type isolate of *A. egyptiacum* (CBS 114785 T) and other isolates of this species (CBS 124.42, A101, A130).

All *Fusarium* isolates successfully grew at a slightly alkaline pH of 8. Incarnatum-Equiseti complex isolates (p7, p12) and the Burgessii complex isolate (p28) turned out to reveal a moderate alkalitolerant phenotype, as they grew best at pH 8 (p7) or in the range 6–8 pH (p12, p28) and had drastically reduced growth at pH 10. Solani complex isolates (p9, p13) were weak alkalitolerants, as their growth slowed down at pH 8 and preferred a slightly acidic medium (pH 6). At pH 10, extra salt in the medium improved the growth of three isolates (p7, p9, p28) (Figure 12a).

As for NaCl preferences, Incarnatum-Equiseti complex isolates (p7, p12) possessed a moderate halotolerant phenotype, as they preferred 2.5% NaCl and grew much slower without salt. The Burgessii complex isolate (p28) was also a moderate halotolerant that preferred 0–2.5% NaCl. One isolate (p9) of the Solani complex was a moderate halotolerant, growing best at 0–2.5% NaCl, while another (p13) was a weak halotolerant, as it preferred a medium without salt, and its growth rate gradually decreased with increasing salt concentrations (Figure 12b).

*A. egyptiacum* isolates were moderate (p19) and weak (p33) alkalitolerants, as they preferred pH 6 or pH 8 (p19) and pH 6 (p33) and had drastically reduced growth at pH 10. Extra salt in the AA medium improved the growth of these strains (Appendix A). Both isolates were moderate halotolerants that preferred 2.5% NaCl (p19) or 0–2.5% NaCl (p33), and their growth rate gradually decreased as the salt concentration increased, though even at 10% NaCl, they grew successfully (Appendix A).

#### 3.2.6. Low-Frequency Isolates from Sediments of Big Tambukan Lake

The isolates that possessed a low frequency of occurrence (i.e., 10%) and occurred in samples of sediments once were *Gibellulopsis nigrescens* (p8), *G. serrae* (p17), and *Chordomyces* sp. (p42) (Plectosphaerellaceae, Glomerellales); *Myriodontium keratinophilum* (Incertae sedis, Onygenales) (p15); and *Pseudeurotium bakeri* (Pseudeurotiaceae, Thelebolales) (p47).

A phylogenetic analysis of Plectosphaerellaceae was conducted using ITS sequences from 3 isolates that were aligned to 12 sequences from GenBank, including 1 outgroup taxon (*Plectosphaerella cucumerina* CBS 137.37 T) (Table 1; Appendix A; Appendix A). Two strains (p8, p17) were identified as members of the *Gibellulopsis* genus. Isolate p8 clustered with the ex-type isolate of *G. serrae* (CBS 290.30 T), while isolate p17 clustered with the neotype isolate of *G. nigrescens* (CBS 120949 NT). Isolate p42, with a high support value (97%), was grouped with representatives of the *Chordomyces* genus.

The sequence similarity searches for isolate p15 performed in GenBank using the BLAST sequence analysis tool showed the closest match (100% similarity) in the ITS locus with *M. keratinophilum* CBS 256.81 (GenBank: MH861337.1). The observed micromorphological characteristics also confirmed the identification.

Isolate p47, obtained as a single isolate from sample no. 10 and grown on nutrient media, exhibited the ability to produce numerous fruiting bodies typical of *P. bakeri* (Appendix A). Our results showed that isolate p47 is most closely related to the ex-type strains of *P. bakeri* CBS 878.71 T and *P. bakeri* CBS 128112, with a bootstrap value of 99% (Table 1; Appendix A; Appendix A).

The listed low-frequency isolates showed different pH preferences. Indeed, Plectosphaerellaceae isolates (p8, p17, p42) were strong alkalitolerants that grew optimally at pH 8 and pH 10 (p8 and p42) or successfully developed at pH 6, pH 8, and pH 10 (p17). At the same time, *M. keratinophilum* (p15) was a moderate alkalitolerant with equally high growth rates at pH 6 and pH 8 and noticeably decreased growth at pH 10, while *P. bakeri* (p47) could not grow at alkaline pH values at all and was the single non-alkalitolerant isolate. The addition of extra salt to the medium at pH 10 slightly increased the growth of Plectosphaerellaceae isolates and did not have a visible effect on *M. keratinophilum* (Figure 13a).

NaCl preferences were different among Plectosphaerellaceae. *G. serrae* (p8) was a weak halotolerant and preferred a medium without salt, while *G. nigrescens* (p17) and *Chordomyces* sp. (p42) were moderate halotolerants with optimal growth on media including 2.5% and 5% NaCl. All Plectosphaerellaceae isolates had drastically reduced growth at 10% NaCl (Figure 13b). Then, *M. keratinophilum* (p15) was strongly halotolerant and grew optimally in an extremely wide range of 0–7.5% NaCl. Finally, *P. bakeri* (p47) was a weak halotolerant. It had significantly decreased growth when the NaCl concentration increased and did not grow at 7.5% NaCl (Figure 13b).

### 3.3. Antimicrobial Activity

The antimicrobial characterization of the fungi from sediments of Tambukan Lake against test cultures is given in Table 3 and Appendix A. The analysis of the antimicrobial activity of different genera against test cultures of fungi and bacteria revealed that the majority of the strains are defined by their antibacterial activity. Thus, the proportion of strains with antifungal activity against *A. niger* INA 00760 was 66.7%, while those with antibacterial activity against *B. subtilis* ATCC 6633 made up 87.5% of the total number of studied strains (Table 3).

As expected, the targets of antimicrobial activity varied between the representatives of the different fungal genera. Representatives of micromycetes from the *Emericellopsis* genus showed both antibacterial activity (i.e., predominantly against *B. subtilis* ATCC 6633) and antifungal activity (i.e., against *A. niger* INA 00760 and *C. albicans* ATCC 2091). It is important to highlight that, in the present study, *Emericellopsis* strains were more frequent than the other isolated fungal cultures. Weak antifungal activity but a strong antibacterial effect was shown for *Penicillium* and *Aspergillus* genera. It was shown that the *Fusarium* genus produced compounds that were effective against fungi but not against *E. coli* ATCC 25922.

Most of the fungal cultures studied exhibited antibacterial activity against *B. subtilis*, with a zone of inhibition from 9 to 25 mm, rather than against *Escherichia coli* (Table 4).

Thirteen strains from the *Emericellopsis* genus were determined to be highly active against *A. niger* INA 00760 (Table 5). Moreover, two highly active strains against *A. niger* INA 00760 were detected among *Acremonum* and *Chordomyces* genera.

Strains of *Emericellopsis* showed the maximum activity against bacteria when cultivated on CZA; the maximum activity against test cultures of fungi was found when cultivated on AA. At the same time, isolates representing other genera showed approximately the same activity against *B. subtilis* ATCC 6633 and *A. niger* INA 00760, both when cultivated on CZA and when cultivated on MYA. The maximum activity against *C. albicans* ATCC 2091 was observed when cultivated on MYA, and that against *E. coli* ATCC 25922 occurred when cultivated on CZA. The results are presented in Appendix A. The peptaibol emericellipsin’s complex was found in two strains from the Terrestrial clade (p26 and p29) and one from the Soda clade (p46).

## 4. Discussion

Nature possesses a wide variety of environmental niches with diverse abiotic conditions. In some locations, abiotic factors (such as ion content and pH) may deviate from those of most habitats. This leads to the formation of specific zones termed “extreme habitats” that restrict the growth of most organisms. It is now widely known that a wide range of extreme-tolerant and a smaller range of extremophile fungi not only survive but also function in these conditions. Such fungi degrade organic matter, form associations with plants, participate in biogeochemical cycles, etc. However, their success depends on the peculiarities of their adaptation to various environmental factors and their combinations. Most data on polyextremophilia concern prokaryotes, while the data on eukaryotes, including cultivated fungi, are scarce. The use of modern methodological techniques for the isolation and analysis of mycobiota shows that, in saline lakes and their surrounding soils, there are mycelial fungi characterized by unique adaptation features, including new taxa [10,14,16,62,63,64].

The first study of cultivated fungi resistant to saline and alkaline conditions from the sediments of the saline and slightly alkaline Tambukan Lake showed their low taxonomic diversity and abundance. All identified fungi were ascomycetes, most of them belonging to Sordariomycetes, Eurotiomycetes, and Dothideomycetes. A high occurrence and abundance were noted for fungi that belong to *Emericellopsis*, *Alternaria*, *Fusarium*, *Aspergillus*, and *Penicillium*. Sporadic *Acremonium*, *Gibellulopsis*, *Chordomyces*, *Pseudeurotium*, and *Myriodontium* were detected. Most isolates were alkalitolerant (except for p47) and halotolerant. No alkaliphiles or halophiles were detected.

The highest occurrence and abundance in the samples of sediments of Tambukan Lake were observed for *Emericellopsis* spp. Although, in general, species of the genus *Emericellopsis* can be isolated from a variety of substrates worldwide, including agricultural and forest soils, peat deposits, freshwater, and estuarine and marine mudflats [65,66], many researchers have noted that aquatic systems may be rich sources of species of this genus [59,61,67,68]. Currently, the genus comprises 27 species [61,66,68,69,70]. Phylogenetic analysis shows the separation of species of this genus into two large, well-supported clades: the Marine clade and the Terrestrial clade [59]. Later, a separate clade, the Soda clade [60], was proposed for strains of *Emericellopsis* isolated from saline soils surrounding inland lakes. This division only partially reflects the characteristic habitat of specific species of the genus *Emericellopsis*. Thus, some species isolated from marine habitats are included in the Terrestrial clade, for example, *E. brunneiguttula*, *E. enteromorphae*, and *E. tubakii* [66,68]. But for most species, a relationship between the habitat and a certain clade could be traced, which makes it possible to predict the resistance of isolated strains to the action of natural stress factors such as salt and pH. Strains of *Emericellopsis* from Tambukan Lake sediments belonging to the Soda (p30, p36, p43), Marine (13 strains), and Terrestrial (p24, p26, p29, p49) clades showed different degrees of adaptation to pH values and salt concentrations.

Species from the Terrestrial clade (including 18 *Emericellopsis* species) are generally found in soils. Some strains from this clade were also detected in estuarine sediments and unusual biotopes: *E. exuviara* from the skin of *Corucia zebrata* (Scincidae) [71]; *E. moniliformis* from forest soil, from volcanic ash soil, and from human nails [66,72]; and *E. fimetaria* from the soil and from the dung of rabbits [66]. This group is characterized by weak pH tolerance [56,60]. Most isolates of *Emericellopsis* from the Tambukan Lake sediments from the Terrestrial clade (*E. fimetaria*, *Emericellopsis* sp.) were moderate haloalkalitolerants. Species of the Marine clade (comprising eight species) are quite often found in saline marine biotopes on a variety of substrates—in deep-sea muds, on the surface of macrophyte algae, and on invertebrate animals [59,60,61,68]. For *E. atlantica* and *E. cladophorae*, the presence in the genome of enzymes capable of degrading algae-containing substrates has been shown [61,73]. However, the species diversity of this clade is probably significantly higher [56,60]. Phylogenetic analysis of *Emericellopsis* from the Tambukan Lake sediments also indicates isolates forming distinct clades within this highly supported group (Figure 3; Appendix A), which, in further studies, can be described as new species. The Soda clade is close to the Marine clade and includes only one species of *E. alkalina*, as well as several anamorphic representatives of this genus, forming separate clades with high support [60]. All *Emericellopsis* isolates from the Soda and Marine clades from the Tambukan Lake sediments were strong or moderate haloalkalitolerants. Some *Emericellopsis* spp. isolates showed greater resistance to high pH when NaCl was present in the medium. The detection of *Emericellopsis* spp. in various biotopes probably reflects not only the adaptive features of species of this genus but also the diversity of enzymes in them that allow the use of a wide variety of substrates on the shores of saline lakes, from cyanobacterial mats to the remains of *Artemia salina* crustaceans and halophytes [60,61,64,73,74]. The investigated sediments from Tambukan Lake included similar substrates. The detection of numerous isolates of *Emericellopsis* spp., which are resistant to the pH and NaCl concentration, indicates their participation in degradation processes.

Widespread in different habitats, *Alternaria* includes species with a variety of tolerances to pH and salt concentrations. For example, the Soda section includes species isolated from soda soils and demonstrable facultative alkaliphiles [16]. *A. alternata*, the only species known to be widely distributed and usually associated with plants, was isolated from the sediments of Tambukan Lake with a frequency of occurrence of 60%. Isolates of the species showed poor resistance to pH above 8 and salt concentrations above 2.5%. In comparison with other studied saline biotopes, the absence of other Pleosporales species here, which are often important subdominants or dominants under similar conditions, was unexpected [16,63,64]. This is probably explained by the absence of plant remains in the samples we studied.

The high occurrence and high abundance of Eurotiomycetes (*Penicillium* spp. and *Aspergillus* spp.) in the sediments of Tambukan Lake was surprising. They showed weak or moderate alkalitolerance and preferred media containing NaCl, and they were strong halotolerants. The greatest salt resistance among all isolated *Aspergillus* is characteristic of the *Flavipedes* series strains, which showed an increase in growth rate at 10% NaCl. Previously, *Aspergillus* spp. from the same series were isolated from the hypersaline Baskunchak Lake [21]. Isolates of these widely distributed genera are found under various saline conditions [75,76,77,78]. Some species serve as models for studying salt and pH adaptations [14]. Our study confirms that the haloalkalitolerance of *Penicillium* and *Aspergillus* is common among different sections and series.

*Fusarium* spp. and *Acremonium* spp. often show persistence over a wide range of pH values [16]. In the sediments of Tambukan Lake, the occurrence of *Fusarium* spp. was 50%. *Acremonium* is represented by a single species, *A. egyptiacum*. This species was also isolated from the hypersaline Baskunchak Lake [21]. Strains from Tambukan Lake had haloalkalitolerance, showing similar growth rates at pH 6 and 8 and reduced growth at pH 10. Some species of Plectospherellaceae are dominant in conditions of high salinity and high pH, for example, in soda lakes [16,63,64]. At the same time, they can be found in salt lakes [21] and in non-saline soils [19,79]. In the sediments of Tambukan Lake, we found only *Gibellulopsis* spp. and *Chordomyces* sp. They are characterized by strong alkalitolerance and weak or moderate halotolerance.

The detection of the keratinophilic fungus *M. keratinophilum* was interesting, with an occurrence of 10%. Although the fungus does not belong to the dominant species, its detection in the sediments should be considered, as there are cases of its indication as an agent of human keratomycoses [80]. In the hypersaline Baskunchak Lake, we also detected, with a small frequency of occurrence, the keratinophilic fungus *Chrysosporium lobatum* [21]. Unexpectedly, *P. bakeri* was detected, which was previously considered confined to cold biotopes [81]; however, it has been detected in the mudflats and salterns of South Korea [12].

The water–salt balance of Tambukan Lake has changed greatly over the past 100 years, which is reflected in the biota of the lake. For example, the salt-loving crustacean *Artemia salina*, which is characteristic of hypersaline water bodies, disappeared [46]. Monitoring of degraders in these changing conditions is a necessary link in the sequence of measures to preserve the natural properties of the lake. However, detection in salty conditions does not yet mean active participation in the community under these conditions [82]. Difficulties arise in separating permanent inhabitants and active participants in destruction processes from “visitors”, whose spores are always present in the environment without a significant influence on the community. Actual participation in the degrader community may be indicated by the fitness of isolates of certain species in the extreme conditions arising in saline lakes. The use of metabarcoding to study the mycobiota of salt lakes would undoubtedly reveal a greater diversity of OTUs that can be found in samples [83,84]. Nevertheless, this does not exclude them but makes it necessary to use methods that allow for isolating these organisms in culture, assessing their adaptive capabilities, and searching for valuable secondary metabolites.

Thus, in the complex of haloalkalitolerant fungi isolated from the bottom sediments of Lake Tambukan, there are groups of species adapted to different degrees of high salt concentrations and pH values of the medium. This may reflect significant fluctuations in the mineralization of the lake, leading to the dominance of different types of fungi in different periods. The diversity of substrates at the bottom of the lake allows the development of both saprotrophes and highly specialized species (for example, keratinophils). It can be assumed that with further desalination of the lake, changes in the fungal community will occur in the direction of the increasing occurrence of *Penicillium*, *Aspergillus*, *Alternaria*, *Fusarium*, and others that are, to a lesser extent, associated with cyanobacterial complexes at the bottom of the lake.

As is known, micromycetes from unusual ecotopes may be of interest as producers of novel biologically active antimicrobial compounds [12,13]. Since there is a minimum of information on the biodiversity and potential activity of micromycetes of these rarely studied biotopes, one of the objectives was to assay each isolate’s antimicrobial activity against a number of test bacteria and fungi in vitro. Interest in alkaline fungi is related to the search for new antibiotic compounds that organisms can make under extreme conditions. In general, about 88% of all isolates used in operation are active against *B. subtilis*, and a practically equal number of isolates show activity against *A. niger* and *C. albicans*—about 67% and 63%, respectively. A smaller proportion of the isolates are active against *E. coli*—40%. The results we obtained reflect the general trend in the search for new antimicrobial compounds in recent years: an acute shortage of antibiotics active against Gram-negative bacteria and mycosis pathogens.

Of particular interest are species of the genus *Emericellopsis* as possible producers of new antibiotics. Previously, there was a boom (50–60 years of the 20th century) in the study of their ability to produce B-lactam antibiotics, particularly cephalosporins [85]. Later, several more new antibiotics were discovered. Since the early 2000s, the ability of fungi in this group to produce peptaibols has aroused interest. The identification of different peptaibol groups from filamentous fungi stimulated research on their isolation and characterization [86,87,88]. Further, the threat posed by antibiotic resistance accelerated research on these peptaibiotics. This resulted in a rapid increase in the number of identified compounds, which in turn demanded efficient data registration, organization, and retrieval methods [89]. Responding to this need, a few databases were developed. Among the antimicrobial peptides, peptaibols nowadays represent the largest group, with more than 1400 compounds summarized in the offline version of the “Comprehensive Peptaibiotics Database” [90] and online Norine databases [91]. At the moment, fungi of the genus *Emericellopsis* are known to have potential as producers of antimicrobial peptaibols. Thus, zervamicins produced by *E. salmosynnemata* [92,93] have intrinsic neuroleptic activity. Bergofungins A and B, produced by *E.donezkii*, and bergofungins C and D, produced by *E. salmosynnemata*, have antifungal activity [94,95,96], and heptaibine and emerimicins that synthesize *E. minima* have antibacterial potency [97,98,99]. Previously, *Emericellopsis alkalina* was reported to produce the unique peptaibols emericellipsins A-E, as well as a novel class of antifungal compounds [24,58]. Our research recently showed that emericellipsins A-E are synthesized not only by isolates of *E. alkalina* but also by other species of this genus, which are also active against *B. subtilis, E.coli*, and *A. niger* [56]. From the information presented in this study, we can offer evidence that the *Emericellopsis* strains obtained from the sediments of Tambukan Lake can also be beneficial from an antimicrobial standpoint. In the current study, in two strains from the Terrestrial clade (p26 and p29) and one from the Soda clade (p43), emericellipsins A-E were detected. At the same time, interesting activity was observed with respect to fungal test strains (the mold micromycete *Aspergillius niger* INA 00760 and the yeast *Candida albicans* ATCC 2091). When cultivated on MYA and CZA, almost all isolates were either inactive or weakly active against fungal test strains, and when cultivated on an alkaline medium, these same isolates exhibited moderate or high antifungal activity. The collection of haloalkalitolerant strains of *Emericellopsis* spp. obtained from Tambukan Lake is of particular interest for the search for producers of new antibiotics from the peptaibol group.

## 5. Conclusions

Mycobiota from the bottom sediments of Tambukan Lake include cultivated salt- and alkali-resistant mycelial fungi, which can be considered an important link between degraders and participants in the initial stages of the formation of therapeutic mud. The abundance and species diversity of haloalkalitolerant micromycetes are low, and all isolated fungi are ascomycetes. The specifics of the structure of the fungal complex reflect the desalination that has been taking place in the lake recently.

The complex differs from that in bottom sediments of soda and hypersaline chloride lakes with a low occurrence of Plectosphaerellaceae representatives, unconditional dominance of *Emericellopsis* spp., and a significant proportion of halotolerant *Penicillium* spp. and *Aspergillus* spp. The isolated fungi are adapted to different degrees to high salt concentrations and medium pH values. Among haloalkalitolerant fungi from Tambukan Lake, there are promising strains for further investigation of antimicrobial activity.

## Figures and Tables

**Figure 1 microorganisms-11-02587-f001:**
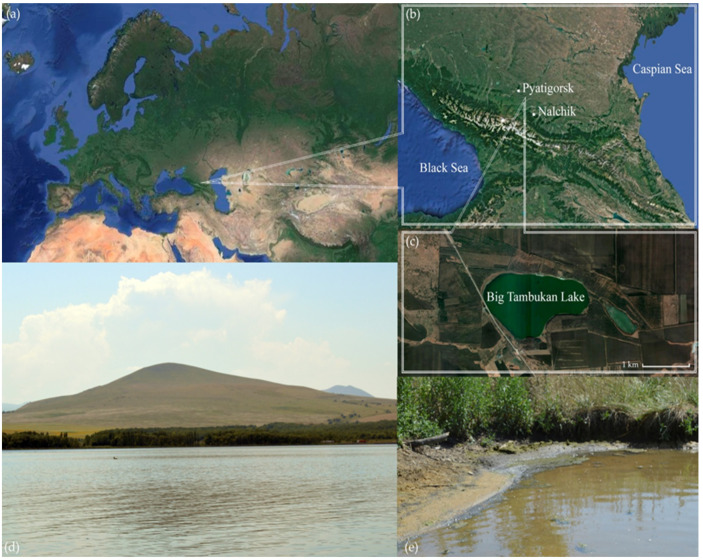
Sampling location. (**a**,**b**)—Location of Big Tambukan Lake; (**c**,**d**)—general view of the lake; (**e**)—view of the sampling place.

**Figure 2 microorganisms-11-02587-f002:**
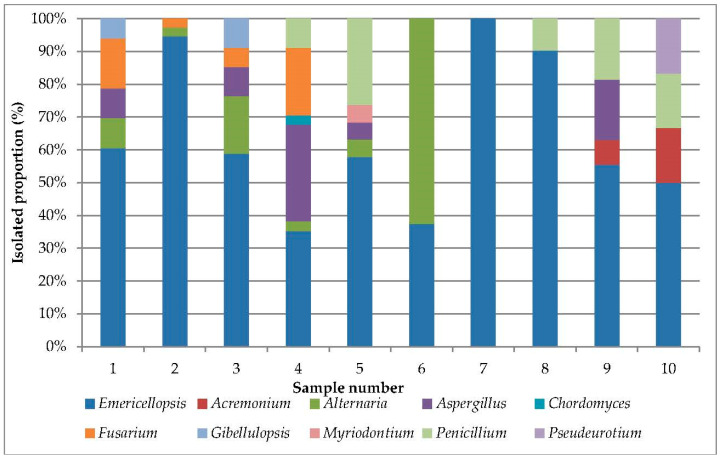
Proportions of filamentous fungal genera in the samples of sediments of Tambukan Lake.

**Figure 3 microorganisms-11-02587-f003:**
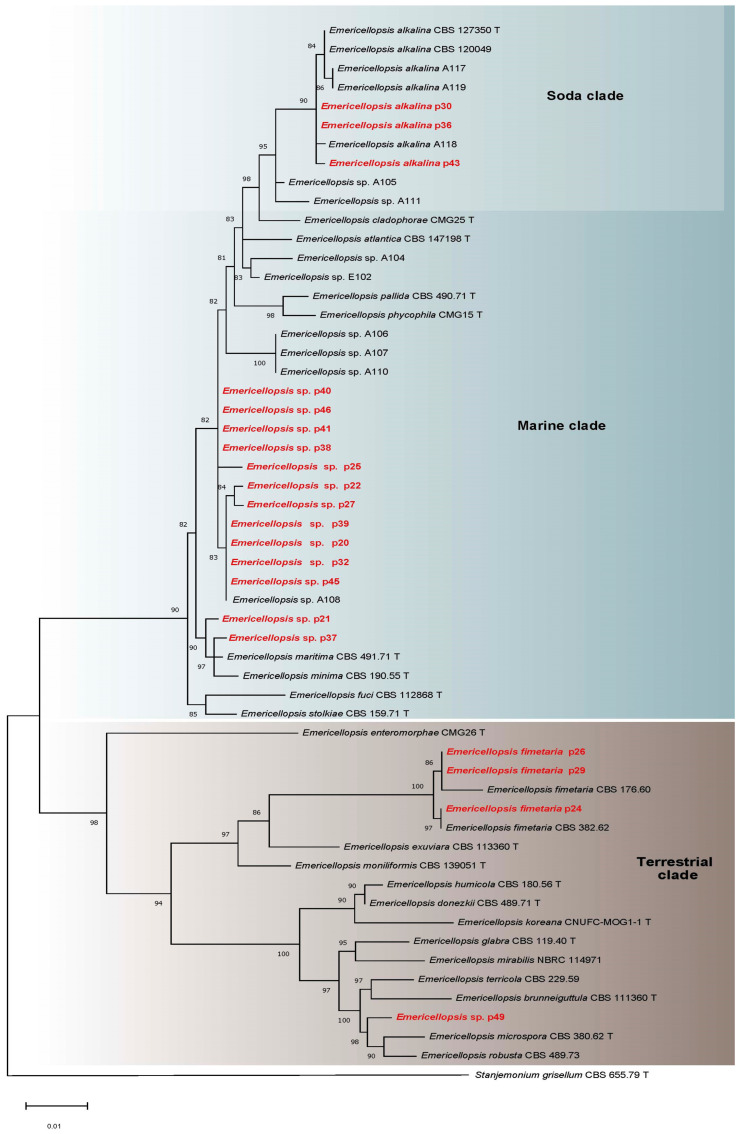
ITS rDNA and b-tub phylogeny of *Emericellopsis* isolates (marked in red bold) and closely related species, with ML support values displayed over each node. *Emericellopsis* spp. and related species were clustered into Marine, Soda, or Terrestrial clades. T—ex-type strains.

**Figure 4 microorganisms-11-02587-f004:**
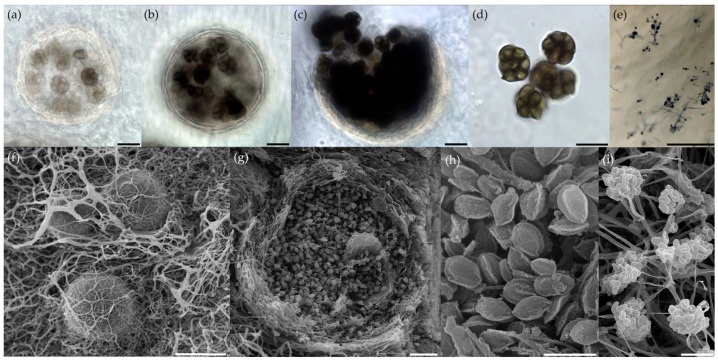
*Emericellopsis fimetaria* p24: (**a**–**c**) early and late stages of ascomata formation (LM); (**d**)—asci with ascospores (LM); (**e**)—conidial sporulation (LM); (**f**)—ascomata covered by hyphal strands with conidiophores (SEM); (**g**)—broken ascomata with ascospores (note multilayered wall) (SEM); (**h**)—ascospores with wings (SEM); (**i**)—conidiophores with conidia (SEM). Scale bars: (**a**–**d**,**f**–**i**) = 10 μm; (**e**) = 100 μm.

**Figure 5 microorganisms-11-02587-f005:**
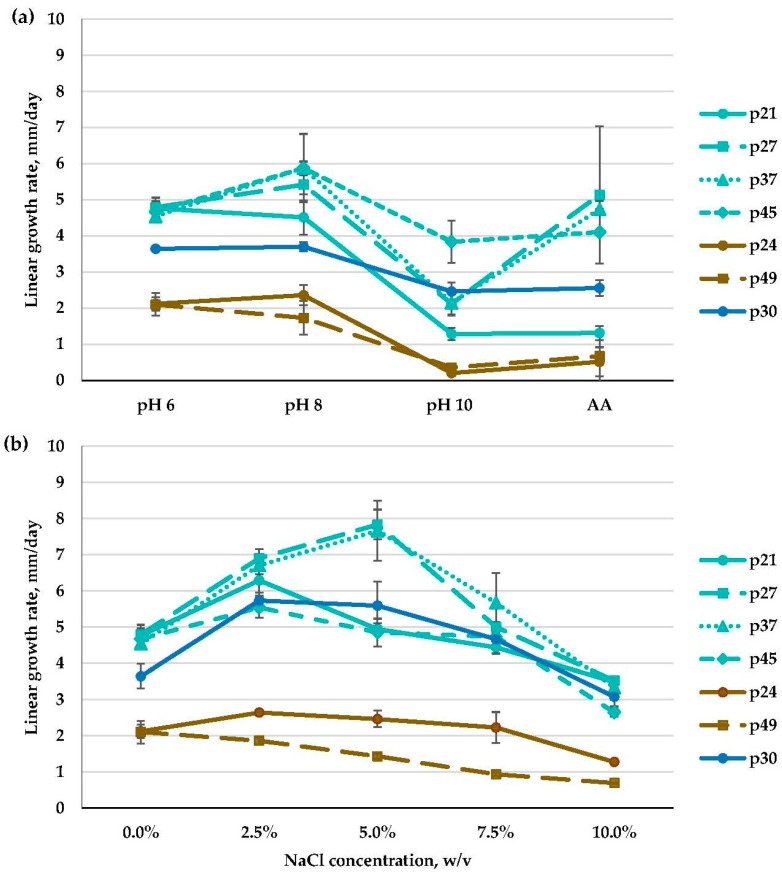
Linear growth rate patterns of *Emericellopsis* strains: (**a**)—at different pH values; (**b**)—at different NaCl concentrations (*n* = 4, α = 0.05, mean ± CI). Soda clade: *E. alkalina* (p30); Marine clade: *Emericellopsis* sp. (p21, p27, p37, p45); Terrestrial clade: *E. fimetaria* (p24) and *Emericellopsis* sp. (p49).

**Figure 6 microorganisms-11-02587-f006:**
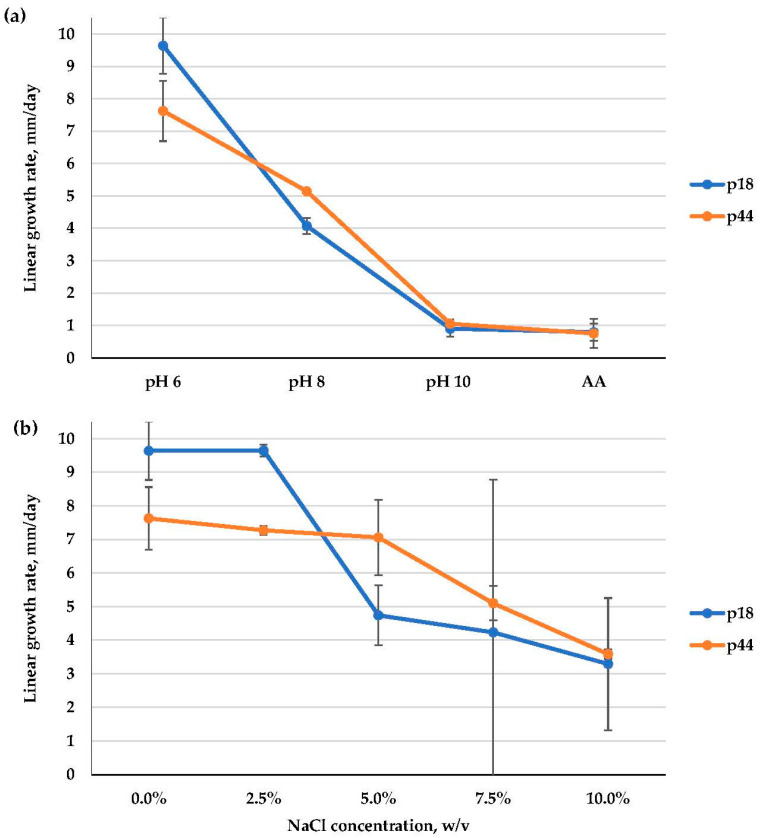
Linear growth rate patterns of *Alternaria alternata*: (**a**)—at different pH values; (**b**)—at different NaCl concentrations (*n* = 4, α = 0.05, mean ± CI).

**Figure 7 microorganisms-11-02587-f007:**
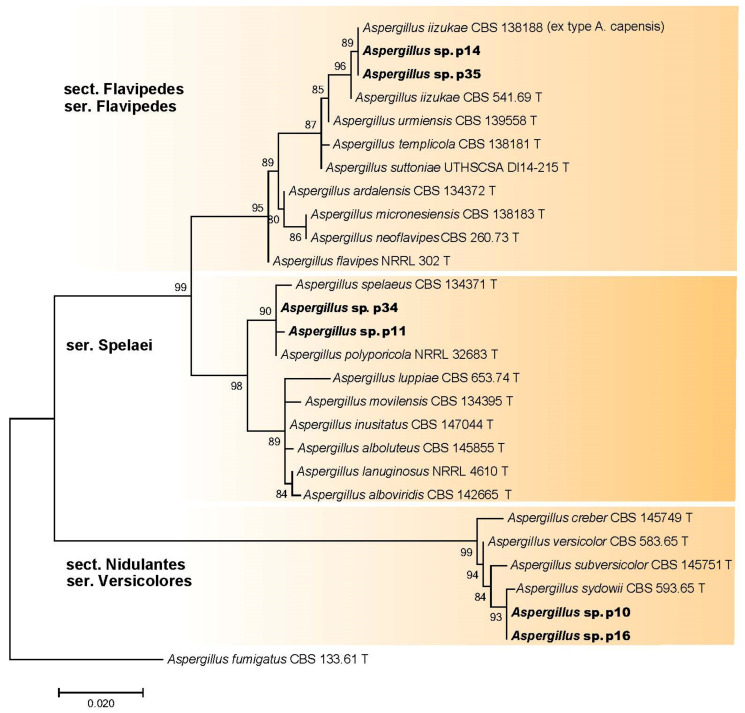
Phylogenetic analysis of *Aspergillus* isolates (marked in bold) and closely related species based on ITS rDNA sequences. ML support values are displayed over each node. T—ex-type strains.

**Figure 8 microorganisms-11-02587-f008:**
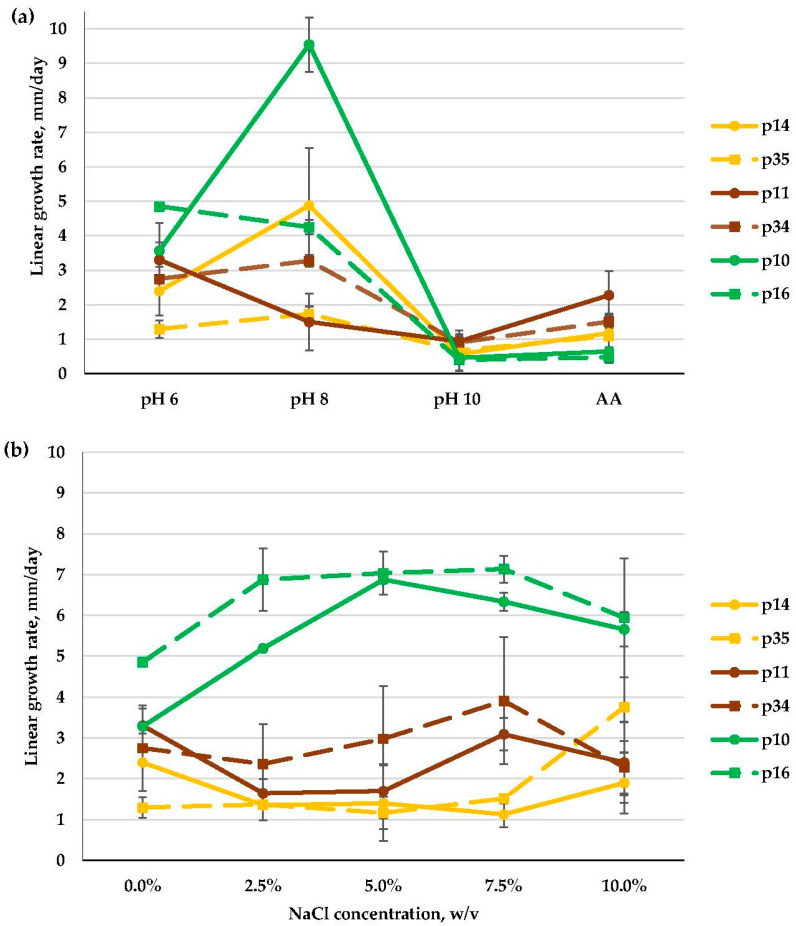
Linear growth rate patterns of *Aspergillus* sp. ser. *Flavipedes* (p14, p35); *Spelaei* (p11, p34); *Versicolores* (p10, p16): (**a**)—at different pH values; (**b**)—at different NaCl concentrations (*n* = 4, α = 0.05, mean ± CI).

**Figure 9 microorganisms-11-02587-f009:**
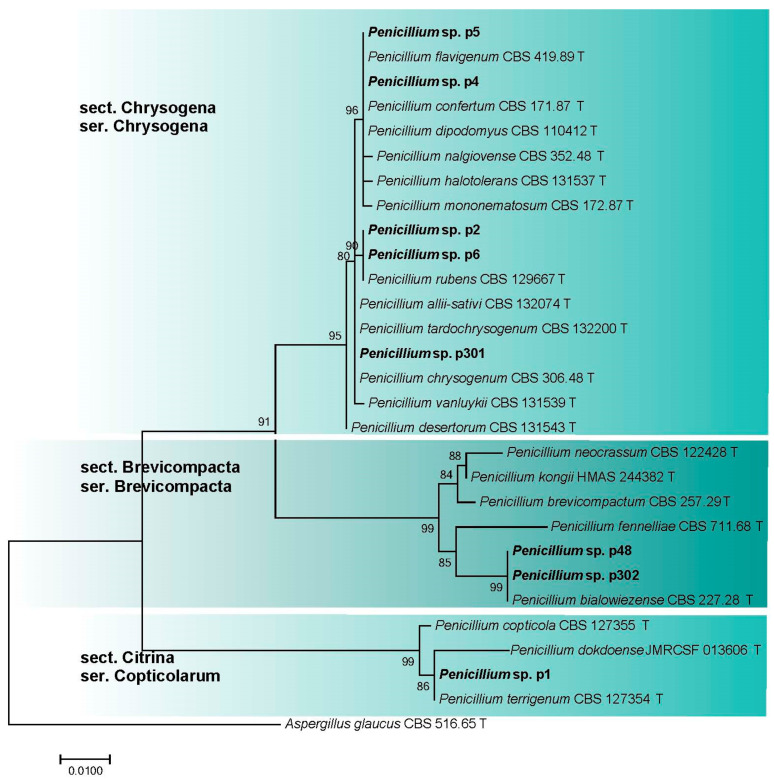
Phylogenetic analysis of *Penicillium* isolates (marked in bold) and closely related species based on ITS rDNA sequences. ML support values are displayed over each node. T—ex-type strains.

**Figure 10 microorganisms-11-02587-f010:**
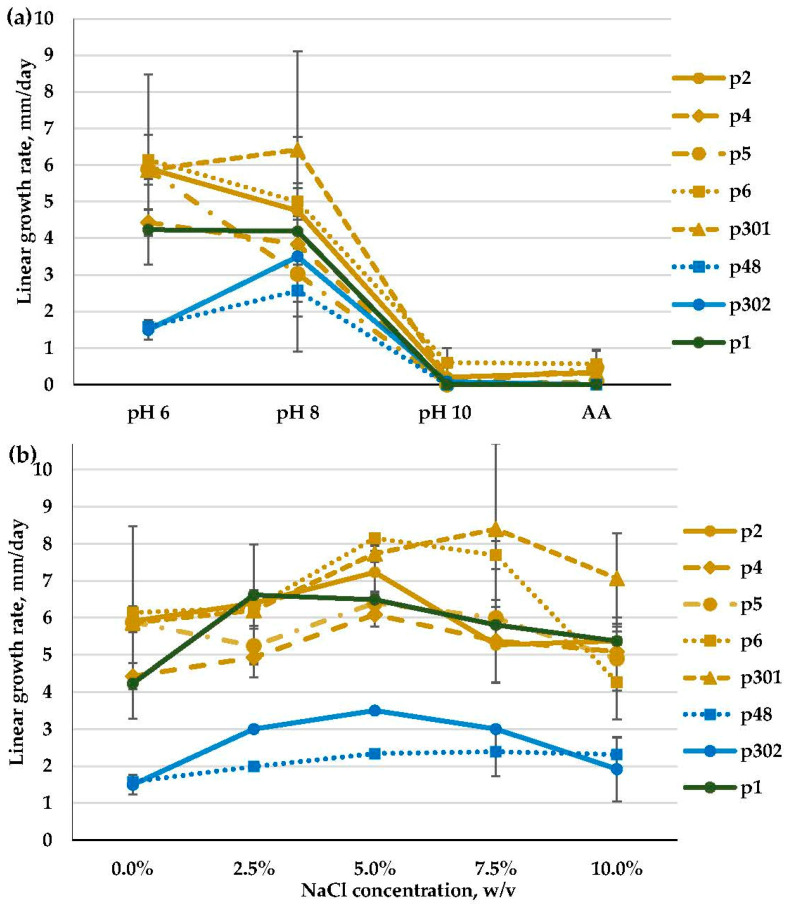
Linear growth rate patterns of *Penicillium* sp. series *Chrysogena* (p2, p4, p5, p6, p301); *Brevicompacta* (p48, p302); *Copticolarum* (p1): (**a**)—at different pH values; (**b**)—at different NaCl concentrations (*n* = 4, α = 0.05, mean ± CI).

**Figure 11 microorganisms-11-02587-f011:**
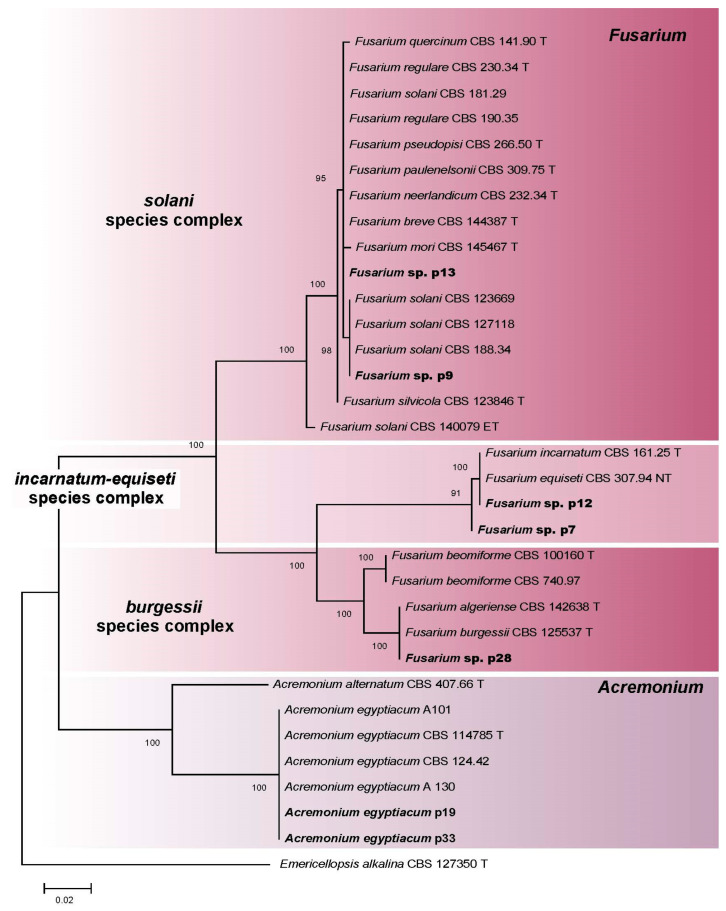
Phylogenetic analysis of *Fusarium* and *Acremonium* isolates (marked in bold) and closely related species based on ITS rDNA sequences. ML support values are displayed over each node. T—ex-type strains; NT—neotype strain.

**Figure 12 microorganisms-11-02587-f012:**
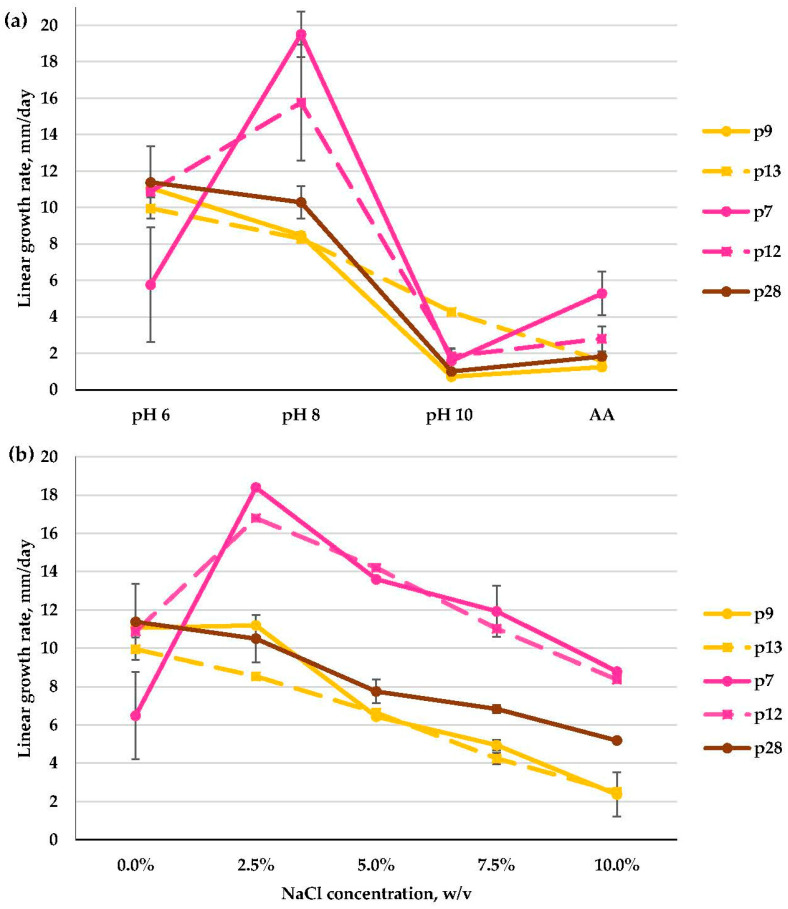
Linear growth rate patterns of *Fusarium* sp. species complexes: Solani (p9, p13); Incarnatum-Equiseti (p7, p12); Burgessii (p28): (**a**)—at different pH values; (**b**)—at different NaCl concentrations (*n* = 4, α = 0.05, mean ± CI).

**Figure 13 microorganisms-11-02587-f013:**
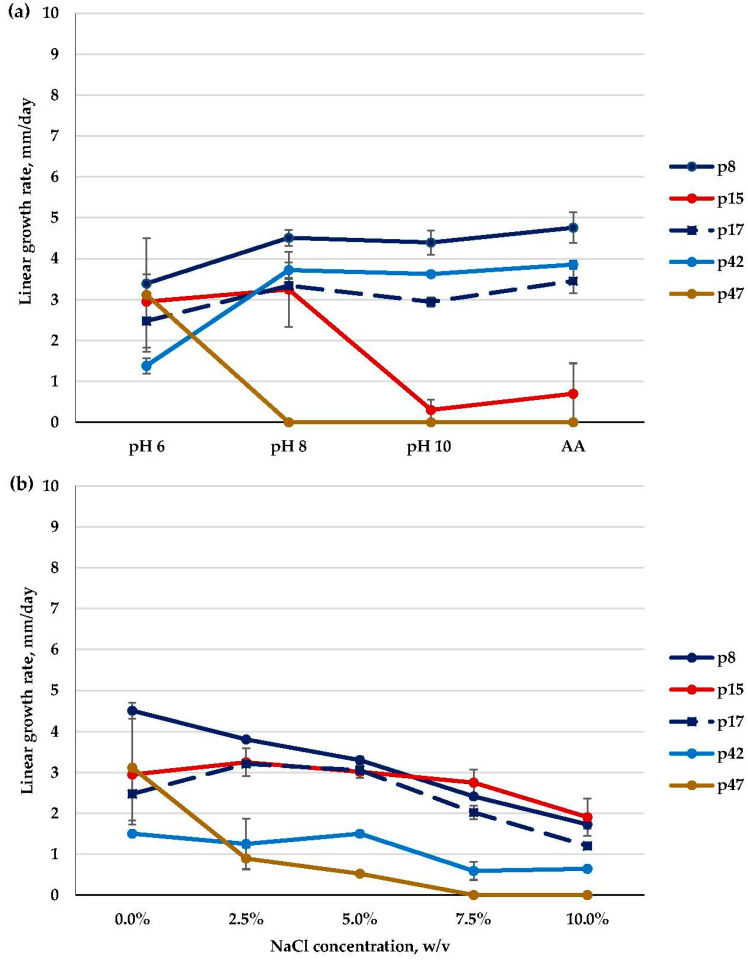
Linear growth rate patterns of *Gibellulopsis serrae* (p8), *G. nigrescens* (p17), *Chordomyces* sp. (p42), *Myriodontium keratinophilum* (p15), and *Pseudeurotium bakeri* (p47): (**a**)—at different pH values; (**b**)—at different NaCl concentrations (*n* = 4, α = 0.05, mean ± CI).

**Table 1 microorganisms-11-02587-t001:** Accession numbers of sequences generated in this study.

Species	Isolate No.	GenBank Accession No.	
		ITS	Beta-Tubulin
*Acremonium egyptiacum*	p19 */SLF 0218.0913 **	OR335864	-
	p33/SLF 0218.1001	OR335865	-
*Alternaria alternata*	p18/SLF 0218.0604	OR335862	-
	p44/SLF 0218.0306	OR335861	-
*Aspergillus* sp. sect. *Flavipedes* ser. *Flavipedes*	p14/SLF 0218.0503	OR335848	-
*Aspergillus* sp. sect. *Flavipedes* ser. *Flavipedes*	p35/SLF 0218.0110	OR335851	-
*Aspergillus* sp. sect. *Flavipedes* ser. *Spelaei*	p11/SLF 0218.0402	OR335847	-
*Aspergillus* sp. sect. *Flavipedes* ser. *Spelaei*	p34/SLF 0218.0911	OR335850	-
*Aspergillus* sp. sect. *Nidulantes* ser. *Versicolores*	p10/SLF 0218.0315	OR335846	-
*Aspergillus* sp. sect. *Nidulantes* ser. *Versicolores*	p16/SLF 0218.0914	OR335849	-
*Chordomyces* sp.	p42/SLF 0218.0408	OR335860	-
*Emericellopsis alkalina*	p30/SLF 0218.0608	OR335874	OR287057
	p36/SLF 0218.0313	OR335876	OR287059
	p43/SLF 0218.1002	OR335882	OR287065
*Emericellopsis fimetaria*	p24/SLF 0218.0620	OR335869	OR287052
	p26/SLF 0218.0308	OR335871	OR287054
	p29/SLF 0218.0609	OR335873	OR287056
*Emericellopsis* sp.	p20/SLF 0218.0117	OR335866	OR287049
*Emericellopsis* sp.	p21/SLF 0218.0708	OR335867	OR287050
*Emericellopsis* sp.	p22/SLF 0218.0101	OR335868	OR287051
*Emericellopsis* sp.	p25/SLF 0218.0601	OR335870	OR287053
*Emericellopsis* sp.	p27/SLF 0218.0504	OR335872	OR287055
*Emericellopsis* sp.	p32/SLF 0218.0908	OR335875	OR287058
*Emericellopsis* sp.	p37/SLF 0218.0401	OR335877	OR287060
*Emericellopsis* sp.	p38/SLF 0218.0203	OR335878	OR287061
*Emericellopsis* sp.	p39/SLF 0218.0701	OR335879	OR287062
*Emericellopsis* sp.	p40/SLF 0218.0702	OR335880	OR287063
*Emericellopsis* sp.	p41/SLF 0218.0312	OR335881	OR287064
*Emericellopsis* sp.	p45/SLF 0218.0201	OR335883	OR287066
*Emericellopsis* sp.	p46/SLF 0218.0801	OR335884	OR287067
*Emericellopsis* sp.	p49/SLF 0218.1006	OR335885	OR287068
*Fusarium* sp. complex Burgessii	p28/SLF 0218.0204	OR335857	-
*Fusarium* sp. complex Incarnatum-Equiseti	p7/SLF 0218.0106	OR335853	-
*Fusarium* sp. complex Incarnatum-Equiseti	p12/SLF 0218.0404	OR335855	-
*Fusarium* sp. complex Solani	p9/SLF 0218.0309	OR335854	-
*Fusarium* sp. complex Solani	p13/SLF 0218.0405	OR335856	-
*Gibellulopsis nigrescens*	p17/SLF 0218.0103	OR335859	-
*Gibellulopsis serrae*	p8/SLF 0218.0302	OR335858	-
*Myriodontium keratinophilum*	p15/SLF 0218.0510	OR335852	-
*Penicillium* sp. sect. *Brevicompacta* ser. *Brevicompacta*	p302/SLF 0218.0562	OR335841	-
*Penicillium* sp. sect. *Brevicompacta* ser. *Brevicompacta*	p48/SLF 0218.1005	OR335845	-
*Penicillium* sp. sect. *Chrysogena* ser. *Chrysogena*	p2/SLF 0218.0902	OR335839	-
*Penicillium* sp. sect. *Chrysogena* ser. *Chrysogena*	p301/SLF 0218.0561	OR335840	-
*Penicillium* sp. sect. *Chrysogena* ser. *Chrysogena*	p4/SLF 0218.0915	OR335842	-
*Penicillium* sp. sect. *Chrysogena* ser. *Chrysogena*	p5/SLF 0218.0903	OR335843	-
*Penicillium* sp. sect. *Chrysogena* ser. *Chrysogena*	p6/SLF 0218.0802	OR335844	-
*Penicillium* sp. sect. *Citrina* ser. *Copticolarum*	p1/SLF 0218.0407	OR335838	-
*Pseudeurotium bakeri*	p47/SLF 0218.1004	OR335863	-

*—working isolate number; **—isolate number in the Collection of Extremophilic Fungi of the Department of Mycology and Algology of Moscow State University.

**Table 2 microorganisms-11-02587-t002:** Haloalkalitolerant ascomycetes isolated from the sediments of *Tambukan* Lake.

Taxon	Frequency of Occurrence, %	Abundance, %
SORDARIOMYCETES, Hypocreales, Bionectriaceae		
*Acremonium egyptiacum*	20	1.1
*Emericellopsis alkalina*	50	3.3
*Emericellopsis fimetaria*	20	1.1
*Emericellopsis* spp. Marine clade	100	61.4
*Emericellopsis* sp. Terrestrial clade	10	0.4
SORDARIOMYCETES, Hypocreales, Nectriaceae		
*Fusarium* sp. complex Burgessii	10	0.4
*Fusarium* sp. complex Incarnatum-Equiseti	20	2.9
*Fusarium* sp. complex Solani	20	2.2
SORDARIOMYCETES, Glomerellales, Plectosphaerellaceae		
*Chordomyces* sp.	10	0.4
*Gibellulopsis nigrescens*	10	0.7
*Gibellulopsis serrae*	10	1.1
DOTHIDEOMYCETES, Pleosporales, Pleosporaceae		
*Alternaria alternata*	60	9.9
EUROTIOMYCETES, Eurotiales, Aspergillaceae		
*Aspergillus* sp. sect. *Flavipedes* ser. *Flavipedes*	30	1.8
*Aspergillus* sp. sect. *Flavipedes* ser. *Spelaei*	20	4.0
*Aspergillus* sp. sect. *Nidulantes* ser. *Versicolores*	20	2.2
*Penicillium* sp. sect. *Brevicompacta* ser. *Brevicompacta*	20	1.1
*Penicillium* sp. sect. *Chrysogena* ser. *Chrysogena*	30	4.0
*Penicillium* sp. sect. *Citrina* ser. *Copticolarum*	10	1.1
EUROTIOMYCETES, Onygenales, Incertae sedis		
*Myriodontium keratinophilum*	10	0.4
LEOTIOMYCETES, Thelebolales, Pseudeurotiaceae		
*Pseudeurotium bakeri*	10	0.4

**Table 3 microorganisms-11-02587-t003:** Total number of strains (%) showing antimicrobial activity.

Genera	Number of Strains	Test Organisms			
		*Escherichia coli* ATCC 25922	*Bacillus subtilis* ATCC 6633	*Aspergillius niger* INA 00760	*Candida albicans* ATCC 2091
*Emericellopsis*	20	13 (65% *)	19 (95%)	18 (90%)	16 (80%)
*Penicillium*	8	2 (25%)	6 (75%)	3 (37.5%)	2 (25%)
*Aspergillus*	6	2 (33.3%)	5 (83.3%)	3 (50%)	4 (66.7%)
*Fusarium*	5	0 (0%)	4 (80%)	3 (60%)	2 (40%)
*Gibellulopsis*	2	1 (50%)	1 (50%)	1 (50%)	0 (0%)
*Alternaria*	2	0 (0%)	2 (100%)	2 (100%)	2 (100%)
*Acremonium*	2	1 (50%)	2 (100%)	1 (50%)	1 (50%)
*Myriodontinum*	1	0 (0%)	1 (100%)	0 (0%)	1 (100%)
*Pseudeurotium*	1	0 (0%)	1 (100%)	0 (0%)	1 (100%)
*Chordomyces*	1	0 (0%)	1 (100%)	1 (100%)	1 (100%)
Total number of strains showing activity	48	19 (39.6% **)	42 (87.5%)	32 (66.7%)	30 (62.5%)

*—Of the total number of isolated strains within the genus. **—Of the total number of strains.

**Table 4 microorganisms-11-02587-t004:** Ratio of the studied strains from different genera with antibacterial activity.

Genera	Total Strains	Inactive		Weakly Active		Moderately Active		Highly Active	
		*Escherichia coli*	*Bacillus subtilis*	*Escherichia coli*	*Bacillus subtilis*	*Escherichia coli*	*Bacillus subtilis*	*Escherichia coli*	*Bacillus subtilis*
*Emericellopsis*	20	7	1	3	0	9	11	1	8
*Penicillium*	8	6	2	0	2	2	4	0	0
*Aspergillus*	6	4	1	1	1	1	4	0	0
*Fusarium*	5	5	1	0	1	0	3	0	0

**Table 5 microorganisms-11-02587-t005:** Ratio of the studied strains from different genera with antifungal activity.

Genera	Total Strains	Inactive		Weakly Active		Moderately Active		Highly Active	
		*Aspergillius niger*	*Candida albicans*	*Aspergillius niger*	*Candida albicans*	*Aspergillius niger*	*Candida albicans*	*Aspergillius niger*	*Candida albicans*
*Emericellopsis*	20	2	4	1	0	4	5	13	11
*Penicillium*	8	5	6	2	2	1	0	0	0
*Aspergillus*	6	3	2	0	1	3	3	0	0
*Fusarium*	5	2	3	0	0	3	2	0	0

## Data Availability

All sequence data are available in NCBI GenBank under the accession numbers in the manuscript.

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
