# Peer review of "Haloalkalitolerant Fungi from Sediments of the Big Tambukan Saline Lake (Northern Caucasus): Diversity and Antimicrobial Potential"

_microorganisms, 2023, doi:10.3390/microorganisms11102587_

Round 1

Reviewer 1 Report

It is always a pleasure for me to read about ecological studies related to extreme environments, however one must be critical when reviewing a manuscript, especially to ensure that the experimental design is appropriate and avoid misinterpretations of the results. A statistical procedure for the data is not reflected in the article. Although the authors refer in some cases to significant differences, they do not give details about the statistical tests used. Furthermore, in antimicrobial activity assays, some controls that are important for this type of assay are not reported.

I consider that the article could be published if the authors correct these details that I kindly suggest.

Author Response

Dear Reviewer 1,

Thank you very much for reviewing the manuscript. We have incorporated the changes suggested by you and have highlighted the changes in the manuscript.

We attach the file with the point-by-point responses to your suggestions.

Thanks in advance, kind regards.

Reviewer 2 Report

The paper is interesting and shows some novelty regarding some of the species isolated and the potential of anti microbial activity of some of the isolates. The title and the method of isolation do not correspond with the experimental approach taken. The authors claim that "the objective...is to provide a description of the diversity...", but they use a selective media for the primary isolation (Alkaline agar medium plus 6g/L of NaCl). This rules out a number of fungi that could be inhabiting the lake but are not able to grow under these conditions. If diversity wants to be explored, media without selection should have been used (MYA with two antibiotics (for example ampicillin and cloramphenicol,) to inhibit bacterial growth would have worked. So, the objective should be rethought and expressed accordingly.

The phylogenetic trees are made only with the ITS region of the large ribososmal subunit (although beta tubulin was also amplified for some isolates, I could not find if concatenated sequences were used to perform more robust  phylogenetic trees). Also, the authors should be careful when using NCBI GenBank retrieved sequences, many of them have not really been verified. In my experience using these sequences leads to misleading classifications. I suggest they use sequences from other data banks which are more reliable, such as MycoCosm.

Specific comments:

Line 41: There are reports of moderate halophile fungi collected from non-exterme environments (see Batista-García, et al. 2014 in PLoS ONE,  or SklenáÅ™ et al., 2017, for example)

Line 76: rephrase this sentence. g/L of what? all the minerals? The word year" should be before the numbers of that year (year 1930)

Line 84: Rephrase the objective: This study only describes limited diversity of alkalitolerant and halotolerant fungi

Line 121: Are the 48 strains really extremophiles? They could be extremotolerant only and depositing such strains in a Extremophile herbarium can lead to mistake and confusion. This is an example of why selecting appropriate sequences for the phylogenetic reconstructions should be carefully selected

Line 101: "d" and "e" are missing in the panels of the figure

Line 128: Was this performed after the 10-21 days incubation? How do the authors discard that many of the colonies could be siblings of faster growing fungi than the original numbers present in the lump samples?

Maybe a strategy of collecting the mud and making serial dilutions seeded in MYA plus antibiotics would have been more reliable

Line 167: Again, Sequences should be retrieved from more reliable data banks such as MycoCosm. Sequences in the GenBank of NCBI are often annotated with only the ITS marker which is poorly reliable for taxonomical certainty

Line 170: Again, did the authors concatenated the sequences to perform the phylogenetic tree? This would result in a much more robust classification

Line 202: authors should homogenize concentration units, sometimes they report %, sometimes g/L. Then it is difficult to compare their results. I would suggest NaCl molarity, most papers report salt tolerance in this units

Line 271-274: Revise the definition of alkaliphylic and alkalitolerant organisms: Then these should be classified as alkaliphylic fungi, since their growth optima is in pH8 rather than in neutral or acidic pH.

The authors should establish the difference between alkaliphyle, alkalitolerant and alkalisensistive fungi according to their optimum growth at different pH

Line 317: Typographical error, the "A" after sp. should be in lower case

Line 371: How much is "small amount"? 6g/L? 

Line 379: Is there an hypothesis for this behavior? This should be discussed in the Discussion section

Line 384: see line 128: p38 and p41 could be siblings of the same fungus: not only they group together in the phylogenetic tree, but show the same behavior regarding halotolerance; they are missing in Figure 5. In the supplementary material it can also be seen that their growth in AA medium is the same

Line 403: According to the growth behavior, these fungi are not even alakalitolerant, their growth is dramatically diminished at pH 8

Line 429: Again, do p14 and p35 could be siblings of the same species?

Please revise carefully these issues for all the phylogenetic analyses and growth behavior for all the species

line 648: and much more, they could associate to the roots of plants seen in figure 1 e? and play a role in the biogeochemical cycles

The whole Discussion section must be modified with the previously exposed ideas. Is there really a low biodiversity in tis site? a metagenome analysis would be more accurate to assess this issue. Even traditional cultivation techniques could give more information about diversity if no selection (i.e. high pH or high salt concentrations) would have been used. 

Also, the phylogenetic analysis is not rigorous, more molecular markers are needed to classify the isolates, especially those belonging to the Emerillopsis genus, Acremonium spp. being the anamorph (asexual state)

In the Conclusions section, lines 820 -821 the sentence is better, since it refers only to the selected fungal communities of the bottom of the lake. This should be reflected in the Title, Introduction, results and Discussion sections.

A deep English revision must be performed, there are many incomplete sentences or wrongly constructed paragraphs. In some cases is difficult to follow what the authors want to mean, there are many sentences that are difficult to follow. (see lanes 68 and 76, for example).

Author Response

Dear Reviewer 2,

Thank you very much for reviewing the manuscript. We have incorporated the changes suggested by you and have highlighted the changes in the manuscript.

We attach the file with the point-by-point responses to your suggestions.

Thanks in advance, kind regards.

Reviewer 3 Report

The manuscript investigates the diversity and antimicrobial potential of haloalkalitolerant fungi from sediments of the big Tambukan Saline Lake (Northern Caucasus). And the results showed that the diversity of haloalkalitolerant fungi was low, however, the majority of the strains were also active against four selected indicator microorganisms. This article presents a wealth of content, including fungal diversity, characterization of cultivated haloalkalitolerant fungi, antibacterial and antifungal activity.

However, I feel that the length of this manuscript is too long to highlight the key content of the manuscript. There was also no discussion on some important topics, such as why the diversity of fungi in these environments is relatively low, and why the proportion of antibacterial or antifungal activity of these strains is so high.

It is recommended that the author abbreviate some of the characteristic descriptions of these strains, or include most of these contents in the supporting materials.

 The specific comments are as follows.

1. Line 118: What is the room temperature?

2. Lines 120: How to select or acquire the 48 strains from these 272 fungal CFU.

3. Figures 3, 6, 8, 10, 12, 15 and 16: The genus and species names of these fungi require italics.

4. Lines 156-158: Numbers and temperature units may be left blank. There are similar issues in other places, please carefully check and correct them.

5. Lines 58, 298, 489 and 799: The article involves the expression of many species, and generally, when a certain species appears for the second time, the genus name should be abbreviated. But in many places, the author did not do so. For examples,  Emericellopsis alkaline in Lines 58, 298, 489 and 799. Please carefully check the similar issues that appear in the article and correct them. There are similar issues in other places, please carefully check and correct them.

Minor editing of English language required

Author Response

Dear Reviewer 3,

Thank you very much for reviewing the manuscript. We have incorporated the changes suggested by you and have highlighted the changes in the manuscript.

We attach the file with the point-by-point responses to your suggestions.

Thanks in advance, kind regards.

Round 2

Reviewer 1 Report

The manuscript has improved considerably, however there are still some errors that need to be corrected.

Author Response

(The authors gave the same response as above.)

Reviewer 2 Report

Again, the main concern in this paper are the phylogenetic reconstructions, which are not convincing. See for example Fusarium, Penicillium and Emericellopsis clades. Look at the Penicillium tree: there are several different "species" that group together in the same clade, this is not consistent with any phylogeny (regardless of the high support values in the branches). The same is true for the Fusarium clade, where also different species are grouped in the same clade. In the Emericellopsis tree also a significant number of isolates group together in a big collapsed branch. A good phylogeny should be able to separate these OTUS as species in different branches

It does not matter that the authors do not have specific primers for all the species analyzed. There are "universal primers" for other molecular markers that usually work for many fungi: RNApolII; ETFT1; Calmodulin... Authors should try to amplify these markers. We have been able to raise PCR products of this markers for Aspergillus and Trichoderma, which are scarcely related phylogenetically, so there is a good chance that they will work for Emericellopsis, Pencinilum*, Alternata, etc.

* I am sure that for Penicillium there must be specific primers for other markers. Probably too for Alternata, since it is a plant pathogen

This also makes uncertain that some of the isolates belong to the same species, regardless that they have been isolated from different Petri dishes or mud samples, they can be abundant species in the Tambukan lake and have grown in different samples.

Maybe a Table showing the different behavior of the isolates grown in alkaline pH and % NaCl could present an argument that they are in fact different species (although this also has its drawbacks). Reading this, in the text is laborious to relate the phenotype with the phylogenetic position of each strain, since one has to refer to the trees and find the number of the species described. Anyway the authors should be careful and state that PROBABLY these strains belong to the assigned genera. In my experience the ITS marker alone is misleading to classify fungi, in fact we had to re-classify a fungus that firs was identified as a certain Aspergillus species and with more markers it turned out to be another. The same happened with a Trichoderma strain.

The English has been improved, but still needs minor revison

Author Response

Dear Reviewer 2,

Thank you very much for reviewing the manuscript. 

We attach the file with the point-by-point responses to your suggestions.

Thanks in advance, kind regards.

Reviewer 3 Report

The author has responded one by one to the questions I am concerned about.

Author Response

Dear Reviewer 3,
Thank you very much for the evaluation of the manuscript. 
Kind regards